# Bioactive Components in Oat and Barley Grain as a Promising Breeding Trend for Functional Food Production

**DOI:** 10.3390/molecules26082260

**Published:** 2021-04-14

**Authors:** Natalia A. Shvachko, Igor G. Loskutov, Tatyana V. Semilet, Vitaliy S. Popov, Olga N. Kovaleva, Alexei V. Konarev

**Affiliations:** Federal Research Center, the N.I. Vavilov All-Russian Institute of Plant Genetic Resources (VIR), 42–44 Bolshaya Morskaya Street, 190000 St. Petersburg, Russia; n.shvachko@vir.nw.ru (N.A.S.); t.semilet@vir.nw.ru (T.V.S.); v.popov@vir.nw.ru (V.S.P.); o.kovaleva@vir.nw.ru (O.N.K.); a.konarev@vir.nw.ru (A.V.K.)

**Keywords:** β-glucans, polysaccharides, flavonoids, anthocyanins, antioxidants, biological role of pigments, gene families

## Abstract

Cereal crops, such as oats and barley, possess a number of valuable properties that meet the requirements for functional diet components. This review summarized the available information about bioactive compounds of oat and barley grain. The results of studying the structure and physicochemical properties of the cell wall polysaccharides of barley and oat are presented. The main components of the flavonoids formation pathway are shown and data, concerning anthocyanins biosynthesis in various barley tissues, are discussed. Moreover, we analyzed the available information about structural and regulatory genes of anthocyanin biosynthesis in *Hordeum vulgare* L. genome, including β-glucan biosynthesis genes in *Avena sativa* L species. However, there is not enough knowledge about the genes responsible for biosynthesis of β-glucans and corresponding enzymes and plant polyphenols. The review also covers contemporary studies about collections of oat and barley genetic resources held by the N.I. Vavilov All-Russian Institute of Plant Genetic Resources (VIR). This review intended to provide information on the processes of biosynthesis of biologically active compounds in cereals that will promote further researches devoted to transcription factors controlling expression of structural genes and their role in other physiological processes in higher plants. Found achievements will allow breeders to create new highly productive varieties with the desirable properties.

## 1. Introduction

Functional food products (FFPs) are gaining more and more popularity on the contemporary healthy food market, since they contain high levels of functional food ingredients (FFIs) which continuous consumption reduces the risks of diseases and metabolic disorders [1]. Such products are intended to be consumed systematically with meals by all age groups of the healthy population. The size of the world’s FFP market in 2018 was estimated at 161.49 billion U.S. dollars, and there are prognoses that it will reach 275.77 billion U.S. dollars by 2025 [2]. Major components of FFPs are three groups of bioactive compounds: prebiotics, probiotics and symbiotics. FFIs are considered to include soluble and insoluble dietary fibers, vitamins, minerals, essential amino acids, unsaturated fatty acids including omega-3 and omega-6, phytosterols, polyols, conjugated isomers of linolenic acid, phospholipids, sphingolipids, and secondary plant compounds (flavonoids, carotenoids, lycopene, etc.) [3,4]. The following basic criteria are used to recognize a product as functional: it should contain only natural substances; it may and must be part of the daily or long-term diet; it should produce a targeted effect on various functions of an organism and possess curative or prophylactic properties. The FFI content in FFPs should not be less than 15% of the daily physiological requirement calculated for one serving of the product [5,6].

Unlike dietary supplements, FFPs are not served as capsules, tablets or powders; a health-friendly ingredient is incorporated directly in the composition of traditional foods at physiological concentrations. Functional foods are conceptualized as food products that underwent elimination, enrichment or substitution in their composition of nutrients (nutritive components, macro- or micronutrients) and bioactive compounds [7]. The ancient aphorism “Let thy food be thy medicine and medicine be thy food”, ascribed to Hippocrates of Kos, still remains vital. The concept opens wide perspectives for the food industry in the context of developing new products with added nutritional value that will be able to contribute to human health [8]. The monitoring performed by healthcare agencies in Russia to assess the population’s diet across different social groups according to their income showed that: (1) a protein deficiency from 15 to 20% of the dietary reference intake was observed among the population groups with low income; (2) a deficiency in the polyunsaturated fatty acids (PUFA) omega-3 and omega-6 under an excessive intake of hard animal fat was found in all population groups; (3) an expressed deficiency in vitamins was typical for more than half of the population, especially in vitamin C (by 70–90%), the B group and folic acid (60–80%), and β-carotene (40–60%); (4) the consumption of cellulose and pectin was almost twice less than the optimal levels; (5) mineral and micronutrient deficiencies were discovered, including 20–55% in iron, calcium, fluorine, selenium, iodine, etc. [9]. In modern interpretation, considering the adopted standards and technical regulations, functional nutrition is the most important element of healthy food, which is duly recorded in regulatory and legal acts. The strategy of food quality improvement adopted in the Russian Federation until 2030 posed as one of the major objectives the provision of safe and high-quality food products to the population in volumes and assortments needed for an active and healthy lifestyle. The raw produce used for FFP usually requires special agricultural practices that must ensure appropriate safety and quality indicators. High production technologies are applied, with ecologically clean and genetically non-modified material [10]. The FFP consumer market is formed with 50–65% of milk products for functional uses, 9–10% of cereal and bakery products, 3–5% of specialized drinks, and 20–25% of other foodstuffs [3]. Cereal crop products, such as bakery and flour confectionary goods, are a promising target for modifications that shape functional properties in food [11].

Naked oats and barley hold particular promise for a variety of FFP, such as flat and tin bread, biscuits, extruded snacks and cereals. Barley bread made from whole-grain flour in various proportions improves glucose tolerance and lowers LDL cholesterol [12]. The growing demand for healthy foods is driving the use of sprouted grains to make functional flour. In a study [13], the conditions of barley grain germination were optimized in order to produce flour with high nutritional and biofunctional properties. Sprouting was shown to significantly increase the content of vitamins B_1_, B_2_ and C, as well as proteins, while the content of fats, carbohydrates, fiber and β-glucans decreased. Total phenolic compounds, γ-aminobutyric acid, and antioxidant activity increased from 2 to 4 times. The study showed that germination at 16 °C for 3–5 days was the optimal process for producing nutritious and functional barley flour. Under these conditions, sprouts retained 87% of the initial β-glucan content, while the levels of ascorbic acid, riboflavin, phenolics, and GABA were 1.4–2.5 times higher than those in non-sprouted grain. Of interest is the work on the use of whole-grain oat and barley for preparing functional drinks, including vegetable milk. Such drinks are rich in vitamins of the B group, complex carbohydrates (starch and non-starch polysaccharides), and minerals. Whole grain used in beverages also contains a large number of various phenolic compounds with antioxidant activity [4]. Supplementing the diet with whole oat grains, rich in β-glucans and arabinoxylans, protects against cardiovascular diseases, type II diabetes, obesity, and some cancers. In a number of countries, such as Finland, the UK or the USA, oat grains have long been used in a gluten-free diet [8,14].

Currently, there are numerous options for the use of β-glucans in food as FFI. They are added to a wide variety of foods, such as baked goods and pasta, muffins, cakes, muesli, dairy products, soups, sauces, drinks, low-fat milk and meat products. At the same time, they have been found to affect the characteristics of food products, in particular, their water absorption capacity, texture, and appearance. By replacing some of the fats in cheeses with β-glucans, it was possible to obtain a softer structure, with a lower melting point and good organoleptic characteristics [15]. Due to its ability to mimic the properties of fats, oat fiber is one of the most effective substitutes to obtain lean meat products, such as beef patties and lean sausages. Breads with oat flavor or taste are extremely popular with customers. The content of oat in bread can reach 50%. With its addition, one can make both wheat bread and rye bread, or all kinds of baked goods. Using whole-grain oat flour obtained from non-toxic oat varieties (cvs. ‘Argamak’, ‘Rhianon’ and ‘Pushkinsky golozerny’), technologies were developed for preparing semifinished profiteroles, wafers and gingerbread products. Foods prepared without sucrose and wheat flour can be recommended for the diet of diabetic and celiac patients. Oat milling products added to bread contribute to an increase in the moisture content, which helps to preserve the freshness of the bread, slowing down the hardening process. This is accomplished by adding high-fiber products, such as bran flour, or pregelatinized oat products [16,17]. Oat-based breakfast cereals are also quite popular on the market. It has been found that the addition of 20% oat β-glucan to the flakes promotes the growth and development of health-friendly intestinal microflora [18,19,20].

Thus, a comparative analysis of a wide range of biochemical characteristics of oats and barley and a study of the molecular genetic mechanisms regulating the biosynthesis of β-glucans, plant polyphenols and other bioactive compounds are of fundamental importance for the development of new cultivars and their further use in breeding practice aimed at obtaining functional food products. The review summarizes information on bioactive compounds in oat and barley grain. The data on regulatory and structural genes of anthocyanin biosynthesis in *Hordeum vulgare* L. and β-glucan biosynthesis in *Avena sativa* L. are analyzed. The review presents modern case studies involving oat and barley accessions from the global plant genetic resources collection of the N.I. Vavilov All-Russian Institute of Plant Genetic Resources (VIR).

## 2. Bioactive Components in Oat and Barley Grain Used in Functional Nutrition

Cereal crops are the most popular natural source of dietary fiber. They contain unique combinations of soluble and insoluble dietary fibers, and polysaccharides, together with low-molecular-weight bioactive components. The main phytochemicals found in cereal crops are phenolic acids, flavones, phytic acid, flavonoids, coumarins, and terpenes. Grain germs are good sources of ferulic and phytic acids, glutathione, and phytosterols. In addition, cereal crop germs contain vitamins E, B_1_, B_2_ and B_3_, minerals P, K, Mg, Ca, Zn and S, and fiber. Due to their rich nutrient content, cereal germs can be a valuable ingredient for FFP production [21].

Oat (*Avena* L.) and barley (*Hordeum* L.) are grain forage crops used for nutritional and dietary purposes. With this in view, quality indicators of grain are becoming increasingly important in the production of these crops, in addition to grain yield [22]. Oat is one of the most promising agricultural crops, since it has a number of valuable properties that meet the requirements for FFP and make it possible to use it this crop for animal feed and for medical or prophylactic purposes. Oat is traditionally regarded as a nutritious cereal crop, contains unsaturated fatty acids, sterols, basic minerals, globular proteins, and β-glucans, and is characterized by the presence of a variety of chemical compounds exhibiting antioxidant properties (tocopherols, etc.) [23,24,25].

Oat and barley globulins are the most balanced in amino acid composition compared with other cereals; specifically, they demonstrate high contents of essential amino acids (arginine, histidine, lysine, tryptophan, etc.) [26]. The protein content in cultivated oats ranges from 12 to 20% [27]. Oat can be grown as a protein crop. At the same time, a protein content of more than 15% was found mainly in naked oat cultivars from the VIR collection, such as ‘Avel’, ‘Mozart’, ‘Numbat’, ‘Sibirsky golozerny’, ‘Persheron’, ‘Vladyka’, and ‘Gavrosh’ [28]. The nutritional value of barley is also associated with the high content of essential amino acids in its protein. Oat grain contains a relatively high amount of oil with a valuable composition. The composition of oat oil includes essential fatty acids (FAs) indispensable for humans: unsaturated ones, such as linoleic (C18: 2, ω-3) and linolenic (C18:3, ω-3), as well as arachidonic acid (C20:4, ω-6), together composing the so-called vitamin F [29]. An important indicator of the nutritional value of oat oil is the content of α-linolenic acid (C18:3), a polyunsaturated omega-3 FA, which plays an important role in the prevention of atherosclerosis [30,31]. A study of the naked oat lines developed in the Volga-Vyatka region showed that the lipid content in their grain varied from 5.91 to 7.87%, averaging 6.9 ± 0.98%. The main FAs of naked oat lipids in the studied lines were palmitic (15.3–17.8%), oleic (33.5–36.7%) and linoleic (36.2–38.7%) acids. According to the content of oleic and linoleic acids and their ratio (1:1), lipids of the naked oat grain belong to the oleic–linoleic group of vegetable oils [32]. Another study [33] resulted in identifying lines of naked oat which received the names ‘Bekas’ and ‘Baget’ after their registration as cultivars in the State Register for Selection Achievements. The content of oleic acid in these cultivars is 36.42 and 33.49%, and that of linoleic acid 35.89 and 38.37%, respectively. The amount of gluten does not exceed 0.2 mg/g. These cultivars can be used for functional and gluten-free food production. In one more study, the oil content varied in oat accessions from 2.7% (‘Skorpion’) to 8.1% (‘Sibirsky golozerny’). Relatively high oil contents (over 6%) were observed in cvs. ‘Avel’, ‘Sibirsky golozerny’, ‘Persheron’ and ‘Vladyka’: 6.6, 8.1, 15.7 and 17.3%, respectively. [28] Wild oats, as a rule, have higher oil contents and the FA ratio of 18:1, generally lower than in cultivated oat (18:2 and 18:3 in FAs). In addition to conventional FAs, a certain amount of hydroxy and epoxy FAs were also present in oat oil, being mostly limited to specific classes of lipids. This study emphasizes the potential of using wild oat species in breeding programs to develop new cultivars of cultivated oat that yield oil with different FA composition and a high FA content [26,34]. Oil and lipid content was compared in wild (spp. *A. fatua*, *A. ludoviciana*, and *A. sterilis*, 6n) and cultivated oats (8 covered: Astor, Lodi, Borrus, Spear, Wright, Fakir, Allyur, Argamak and 2 naked: Torch and Kynon). The oil content of cultivated oats was significantly lower with compared to wild oats, but percentage of minor lipid classes was significantly higher in cultivated accessions (Table 1) [34].

In cereals (unlike most crops), the cell walls of the grain endosperm contain very little cellulose and consist mainly of arabinoxylans and (1,3;1,4)-β-d-glucans which ratio varies significantly across different species: arabinoxylans dominate in rye and wheat, while (1,3;1,4)-β-d-glucans in barley and oats [35,36]. Among cereal crops, the highest content (g per 100 g of dry weight) of β-glucan is observed in barley (2–20 g, 65% of water-soluble fractions) and oat (3–8 g, 82% of water-soluble fractions) [37,38]. Other cereals also contain β-glucans, but in much smaller amounts: 1.1–6.2 g in sorghum, 1.3–2.7 g in rye, 0.8–1.7 g in maize, 0.3–1.2 g in triticale, 0.5–1.0 g in wheat, and 0.13 g in rice [39,40]. Βeta-glucans belong to dietary fibers–high-molecular-weight carbohydrates of plant origin which produce a beneficial effect on important functions of the gastrointestinal tract and systemic processes in the human organism [41]. They help to reduce the risk of cardiovascular diseases, maintain or decrease the amount of blood cholesterol, including low-density one, mitigate the risk of hyperglycemic syndrome, improve liver functions, and reduce excessive body weight [42,43,44,45,46]. It is also believed that insoluble oat fiber reduces the amount of carcinogens in the gastrointestinal tract [47]. The U.S. Food and Drug Administration recommend a daily intake of at least 3 g of β-glucans from oat or barley. The European Food Safety Association has also confirmed the value of β-glucans. It has been established that the water-soluble dietary fibers β-glucans and the phenolic alkaloids avenanthramides may be included into the daily diet as FPIs [48]. Oat grains contain unique antioxidants-avenanthramides, which inhibit inflammatory processes in endothelial cells. It had shown that oat-based diets in mice reduce the risk of atherosclerosis. The processes of lipoes distribution and the becoming of atherosclerosis in mice and humans are liking, therefore oat avenanthramides are very important for the prevention of human cardiovascular diseases [49].

In recent years, there has been an increased interest in the use of antioxidants for treating or preventing diseases associated with oxidative stress [50]. Oat and barley grains are a valuable source of phytoestrogens, vitamin E, and phenolic antioxidants, possessing biological activity and capable of significantly increasing the nutritional value of products made from these grains. Among all cereal crops, γ-tocotrienol (one of the isomers of vitamin E) was found only in barley grain [51]. The presence of these and many other bioactive compounds in oats and barley makes them indispensable products both for patients suffering from various metabolic disorders and for healthy people. It should be noted that, although cereal crops are considered one of the main components of human nutrition, their oxidant activity has not been analyzed profoundly enough. In the case study of 30 different commercial breakfast cereals, it was shown that polyphenol levels in an average serving of oat-based cereals are comparable to those found in an equivalent amount of vegetables or fruits [52]. Another important group of compounds is avenanthramides. These are phenolic compounds with antioxidant, anti-inflammatory, and other types of activity.

## 3. Regulatory and Structural Genes for the Biosynthesis of Anthocyanins in Barley and β-Glucans in Oats

### 3.1. Regulatory and Structural Genes for Anthocyanin Biosynthesis in Hordeum Vulgare

Most researchers are attracted by the genetics of secondary metabolite biosynthesis. Their interest is evoked by the fact that plants synthesizing polyphenolic compounds are promising and, more importantly, readily available phytopreparations. It is already known that flavonoids, such as anthocyanins and proanthocyanidins, possess antioxidant and anti-inflammatory properties, and contain a large amount of vitamins (vitamin P). These valuable features make them indispensable components of a balanced human diet.

A large amount of polyphenolic compounds is found in leaves, flowers, fruits, sprouts, and cover tissues that perform protective functions [53]. Anthocyanins, along with chlorophyll and carotenoids, confer a variety of colors to fruits and seeds and produce a photoprotective effect [54]. Some of them protect plants from pathogenic microorganisms [55]. In addition to the abovementioned functions, this class of compounds provides resistance to limiting and stress factors of biotic, abiotic and anthropogenic nature, exerting a direct impact on plant development [54].

In monocotyledonous plants, flavonoid synthesis is regulated by the MBW protein complex, which includes three groups of regulatory factors: MYB, MYC (bHLH), and WD40. The MBW complex regulates the anthocyanin synthesis process, beginning with synthesis of chalcones up to the appearance of anthocyanin coloration [54] (Figure 1).

Searching for and mapping structural and regulatory genes that control the pathways of biosynthesis and pigment accumulation remain a prioritized task. The main objects of research are representatives of the *Poaceae* Barnnhart family, widely cultivated in Russia and abroad. Among cereal grasses, the formation of flavonoids was studied in more detail in *Hordeum* L. [55,56,57,58,59,60,61], and *Triticum* L. [62,63], while in representatives of the genus *Avena* L. this aspect is still unexplored.

*H. vulgare* is also actively examined for molecular genetic mechanisms regulating the interaction of genes involved in the synthesis of plant polyphenols and those which expression forms a protective response to the effects of pathogenic organisms. Karre et al. demonstrated the relationship between the transcription factor HvWRKY23, which protects barley from the pathogenic *Fusarium* fungi, and the key *CHS* and *DFR* genes. It was found that a knockdown of *HvWRKY23* reduced the expression of key genes for the biosynthesis of hydroxycinnamic acid and flavonoids. As a result, the content of secondary metabolites decreased and the number of ears afflicted by fungi increased [64]. Earlier, the same group of researchers discovered the *HvCERK1* gene which expression affected the vital activity of pathogenic fungi in barley and reduced the expression level of key flavonoid genes [65].

During the synthesis of flavonoids, secondary metabolites are formed–anthocyanins and proanthocyanidins, pigmenting the vegetative and generative parts of the plant in red, blue or purple colors. The first works dedicated to the molecular genetic mechanisms inducing the formation of this group of pigments were carried out by Jende-Strid [56] and Meldgaard [57]. Employing a set of barley mutants, the researchers studied in detail the genes of the *Ant* group: *Ant13*, *Ant17*, *Ant18*, and *Ant21*. According to Jende-Strid, *Ant13* is a regulatory gene that controls the work of structural genes involved in the synthesis of flavonoids [56]. Products of *Ant13* act as transcription factors (TFs) of the *Ant18* gene. In its turn, *Ant18* encodes DFR (dihydroflavonol-4-reductase). The structural *Ant17* gene (3H) encodes flavanone-3-hydroxylase (F3H) [56,57]. The *Ant21* gene expression affects the formation of proanthocyanidins and anthocyanins [56]. Mutations in the *Ant13*, *Ant17* [57], *Ant18* and *Ant21* genes cause the loss of pigmentation in plants [56]. These studies set the vector for further research into the molecular genetic mechanism responsible for the biosynthesis of plant polyphenols–anthocyanins.

By now, researchers have found and studied the sequences of structural genes for anthocyanin coloration in the vegetative and generative parts of *H. vulgare* plants. The gene families involved in the synthesis of flavonoid pigments have been identified for monocotyledonous and dicotyledonous plants. The main role in the synthesis of anthocyanin coloration is played by the enzymes: chalcone synthase [66], chalcone isomerase, [67,68,69,70] flavanone-3-hydroxylase [57,61,71] flavonoid 3′,5′-hydroxylase [61], and dihydroflavonol-4-reductase [67,70].

Regulatory genes encoding bHLH/Myc transcription factors are *HvMyc1* (*Ant2*) and *HvMyc2*. A genome-wide association analysis of two barley populations, with anthocyanin coloration (cv. ‘Retriever’) and without it (cv. ‘Saffron’), revealed the *Ant2* gene. The *Ant2* gene is localized on chromosome 2HL and regulates the accumulation of purple pigments in the pericarp [72]. The *HvMyc2* (4HL) gene provides for the formation of blue and pink anthocyanins in the aleurone layer [59,72]. Himi and Taketa (2015) identified the R2R3-MYB-encoding gene *HvMpc1-H1* (*Ant1*), located on the short arm of chromosome 7H, which determined the color of the barley kernel pericarp [73]. Strygina and Khlestkina (2019) identified the regulatory genes *HvMpc1-H2* and *HvMpc1-H3* that controlled the synthesis of anthocyanins and the color of the aleurone layer. Localization of genes was observed on 4HL [74]. The identification and analysis of the genes encoding TF of the WD40 family and determining the anthocyanin coloration of the kernel were carried out by Strygina et al. (2019). The *HvWD40-1* gene was identified on the long arm of chromosome 6HL. The coding sequence for *HvWD40-2*, paralogous to *HvWD40-1*, was identified on the short arm of chromosome 6HS. The study suggests that it is *HvWD40-2*, together with the coding gene *HvMpc1-H3* (family R2R3-Myb) and *HvMyc2* (family bHLH), that forms the MBW regulatory complex which controls pigmentation of the aleurone layer in barley. No polymorphism was found in the studied genes [75].

Thus, the varied coloration in plants is induced by a large number of structural and regulatory genes responsible for the synthesis of flavonoid compounds. At present, thanks to the accomplishments of leading Russian and foreign researchers, the main components of the flavonoid formation mechanisms have been discovered, and data were obtained on the synthesis of anthocyanins in various tissues of barley. Further study of transcription factors, gene expression and their relationship with other physiological functions of plants will make it possible to develop new barley cultivars with desired properties. Such cultivars will be an integral part of the human diet and will serve as the basis for food security.

### 3.2. Genes for the Biosynthesis of β-Glucans in Barley and Oats

A characteristic feature of plants within the Poaceae family is the presence of (1,3;1,4)-β-d-glucans (β-glucans) in the plant cell walls. The structure of cereal β-glucans consists of unsubstituted unbranched polysaccharides that contain monomeric residues of β-d-glucopyranosyl (Figure 2). The degree of polymerization of cereal β-glucans is about 1000 or more; with this in view, (1,4)-bonds are usually more common than (1,3)-bonds, the ratio of (1,3)-bonds to (1,4)-bonds being usually 1:2, with the exception of *Hordeum vulgare* L. and *Avena sativa* L. in which the (1.3) to (1.4) ratio is 2:1 or higher [76]. For example, this ratio in the water-soluble β-glucan of barley usually ranges from 2.2:1 to 2.6:1 [77]. Beta-glucans accumulate in the cell walls of growing vegetative tissues, occur in the secondary walls of the vascular network, and are the main components of the endosperm walls in cereal grains. They are a source of dietary and functional food. Thus, β-glucans of barley and oats affect the quality of flour, are able to reduce serum cholesterol levels in patients with hypercholesterolemia, and modulate the glycemic index in diabetic patients [37,78].

Synthases encoded by the extensive CESA (cellulose synthase), CSL (cellulose synthase-like) and GSL (glucan synthase-like) gene families are involved in the synthesis of most β-linked polysaccharides in the cell walls of plants belonging to the Poaceae family. The CESA genes encode cellulose synthases, [79] GSL genes encode (1,3)-β-d-glucan synthase (callose synthase), [80,81] and CSL genes encode enzymes that synthesize various β-linked polysaccharides of non-cellulose origin. CSL genes are regarded as candidate genes for enzymes encoding β-glucans [76,82,83]. This gene family was subdivided into 8 gene groups designated CSLA to CSLH. The CSLF and CSLH groups were found only in cereals, [82,84] and the CSLF group was identified as a candidate for genes encoding β-glucans in cereal crops [78,85,86]. While studying the rice genome, six OsCslF genes belonging to the CSLF group were identified on chromosome 7, next to the *Bmy2* marker. The genes were designated as *OsCslF1*, *OsCslF2*, *OsCslF3*, *OsCslF4*, *OsCslF8* and *OsCslF9* [84,87]. Burton et al. [87] identified three markers (*Adh8*, *ABG019* and *Bmy2*) that are significantly associated with β-glucan regulation in barley. One of the three significant marker sequences showed homology with the *CslF* genes in rice [88]. Burton, et al. mapped the *HvCslF* genes for barley and showed that at least two of these genes are mapped in the region of barley chromosome 2H defined by the QTL of (1,3;1,4)-β-glucan close to the *Bmy2* marker.

Using a genome-wide association analysis of three groups of elite oat cultivars, Fogarty et al. detected the *AsCslF6* gene, encoding the synthesis of oat β-glucan synthase. Unlike diploid barley, oat is hexaploid and has three subgenomes: A, C and D. Subgenome-specific expression of homeologues A, C and D of the *AsCslF6* gene showed that *AsCslF6_C* made the least contribution to β-glucan biosynthesis, while oat samples with a low β-glucan content demonstrated the highest input of the *AsCslF6_C* expression. At the same time, multiple homeologous copies of the *AsCslF6_A* and *AsCslF6_D* genes make a significant contribution to the common phenotype of samples with high β-glucan content [86].

Thus, although the structure and physicochemical properties of cell wall polysaccharides in barley and oat plants have been studied in detail, the enzymes and coding genes responsible for their synthesis remain underexplored. Further research into the genetic regulation of β-glucan biosynthesis would contribute to a promising trend in barley and oat breeding, since a high content of β-glucans in cultivars is important for functional food production.

## 4. Study of Oat and Barley Accessions from the VIR Global Collection for the Content of Bioactive Components in Grain

In the global germplasm collection of VIR, among the cultivated oats and barleys, there are numerous naked forms. The naked barley contains more than 13.56% of protein, which is superior to the hulled barley (9.09%) in its nutritional value [89]. The effect of presowing treatment of oat seeds with succinic acid on the yield and quality of green biomass and grain was studied by VIR researchers [90]. Oat accessions were grown at Pushkin Experiment Base of VIR. The results showed that after treatment the quality and yield of green biomass and grain increased. The greatest increase in protein (by 5‒8%) and oil (by 10‒13%) was observed in cv. ‘Astor’ (the Netherlands). The maximum increase in seed yield after treatment was demonstrated by cvs. ‘Hadmerslebener AG’ (by 17‒18%) and ‘Borrus’ (by 9‒13%). Besides, it was noted that the presowing treatment of oat seeds with a solution of succinic acid contributed to a decrease in grain husk content [90]. EC Regulation 41/2009 included oats in the list of gluten-free ingredients safe for celiac disease (chronic intolerance to gluten, i.e., gluten proteins found mainly in wheat, rye and barley kernels), provided that the gluten content in the kernels should not exceed 20 ppm. [91]. The research carried out at VIR showed an important role of oat as a substitute for wheat in the gluten-free nutrition (diet) [92,93].

Using a combined HPLC and LC‒MS analysis, comprehensive data on the total content and composition of avenanthramides were obtained for 120 accessions of cultivated oat and 32 of wild oats from the VIR global collection. An accession of the wild hexaploid species *A. sterilis* L. had the highest total content of avenanthramides in its grain (1825 mg/kg), and among the cultivated oat accessions the highest content (407 mg/kg) was observed in the naked cultivar ‘Numbat’ (Australia) (Table 2) [94].

The study of the mineral composition of various oat varieties from the VIR collection is presented in publications [28,95]. The ranges of micronutrient content in oat grain were as follows: 19–37 mg/kg of Fe, 10–70 of Zn, and 3.5–9.9 of Mn, i.e., 7.0-fold variation in Zn and almost 3-fold variation in Mn were observed [95]. It was shown that such oat accessions as cvs. ‘Boog’, ‘Circle’, ‘Vladyka’ and ‘Gavrosh’ contained the highest amounts of Fe, Mn and Zn [28]. The ranges of micronutrient content in barley grain were as follows: 24–79 mg/kg of Fe, 6–33 of Zn, and 7–21 of Mn. Thus, barley genotypes demonstrated a 3–5.5-fold variation in the content of Fe, Zn and Mn, while genotypic variations in seed micronutrients among wheat and rye cultivars were relatively small (1.5–2 times) [95]. Further search for cultivars rich in mineral composition seems to be an important task in the development of FFP and improvement of micronutrient-based diets.

A study aimed at identifying biochemical differences (metabolite markers) among naked and hulled oat cultivars for subsequent phenotyping the varietal gene pool of common oat was accomplished by Loskutov et al. [96]. The naked oat forms were found to have a higher content of hydroxybenzoic acids, and the hulled forms contained more phenols. It seems interesting to compare metabolomic patterns in the grain of wild species and cultivars from the VIR collection in order to identify potential sources of biochemical quality traits among wild oat species in the breeding process. Metabolites were detected, which content changed in the process of domestication or in which wild oat species differed from the cultivars of this crop. Among these compounds, along with such well-known components of healthy nutrition as oleic acid, glucose, fructose, etc., monoacylglycerols were identified: MAG 16:0, MAG-2 18:2 [97], tocopherols, sterols etc. Content of tocopherols and sterols was studied in oats accessions of VIR collection (Table 3).

The amino acids content was higher and the content of fatty acid was lower in cultivated species compared to wild ones. The AB genome tetraploid wild species were characterized by improved values of fatty and organic acids, sterols, polyhydric alcohols and monosaccharides, but the maximum content of total sugars was found for genome C diploids wild species. However, most of parameters studied were minimum for wild tetraploid species with the AC genome. (Table 4) [97].

The content of β-glucans in barley grain is determined by both the plant genotype and the growing conditions [98,99,100]. Some authors believe that the genotype is of decisive importance, [101,102] while others favor the environmental conditions [103,104]. All biochemical parameters of the caryopsis depend on the growing conditions, but the content of protein, oil and other components, largely depends on the genotype and variety. At the same time, the content of β-glucans is more variable depending on the genotype and variety of the grain, compared to other biochemical parameters. Apparently, the content of β-glucans in naked forms of oats and barley is significantly higher than in hulled varieties. Also, the content of β-glucans in two-row barley varieties is higher than in six-row ones, but the waxy naked grain contains the largest amount of soluble β-glucans, this also depends on the color of the grain hulls. There is a persistent positive or persistent negative relationship between different indicators of grain and β-glucans, such as oil, yield, nature and weight of 1000 grains (unpublished data). A study of 33 cultivars and lines of barley in two arid areas in the United States showed that the variability in the β-glucans content in grain depended on the genotype to an extent of 51% [102] to 66% [104]. At the same time, the effect of environmental conditions on protein content in grain amounted to 69%, and on the yield and test weight of grain to 83 and 70%, respectively [102]. A study of 9 cultivars of barley and 10 of oats ascertained that varietal differences in the content of β-glucans persisted over the years [101]. It was shown that the phase of plant development also affected the β-glucan content in grain. The amount of β-glucans was observed to gradually increase in the process of grain development and reach a plateau or decrease during the maturation period [105]. Currently, there are conflicting data on the relationship between the accumulation of β-glucans in barley grain and the values of 1000 grain weight, protein and starch content [100,106]. Some authors see no interrelation between these features, while others point to a positive correlation. When studying the content of β-glucans in the grain of cultivated six-row and two-row barleys, no differences were found between these groups of cultivars [99]. Contradictory data were also obtained by the studies of naked and hulled barley cultivars. Some authors did not reveal significant differences between these forms, [106,107] while others found that the naked barley had a higher content of β-glucans than the hulled one [99,108]. At the same time, a group of Tibetan naked barleys was found to have the highest content of β-glucans in their grain [98].

VIR has all the possibilities (genetic sources and donors, direct and indirect methods for assessing grain material, etc.) to develop barley cultivars with the following properties: low gluten content (the first high-yielding accessions of cultivars capable of diversifying the diet of celiac patients were obtained through breeding,); increased or decreased β-glucan content; low phytin content or high phytinase activity; and high content of anthocyanidins (e.g., with purple-colored grain) [109].

Protein content is one of the important quality features of both feed and food barleys. Many years of research have shown that the protein content in grain is determined by the genotype, despite the phenotypic variability of the trait. High productive with high-protein content forms were identified among both naked and hulled accessions of barley [110].

A strong impact on the quality of grain is exerted by diseases, therefore the breeding for quality is closely linked with the breeding for disease resistance. The most promising, cost-effective and safe way to reduce grain infestation is to develop cultivars possessing genetic resistance. One of the most widespread cereal crop diseases is *Fusarium* ear blight caused by a set of fungal species within the genus *Fusarium* [111]. Especially dangerous is the contaminated grain, because during the life cycle of *Fusarium* fungi it accumulates secondary fungal metabolites–mycotoxins. Secondary metabolites of this group of fungi have a negative effect on the quality of the produced grain and can cause severe intoxications in farm animals and humans. Despite the widespread incidence of *Fusarium* in Russia, practically no breeding efforts are being made to develop resistance to this dangerous disease in barley. According to the results of a long-term assessment of 60 local varieties, improved cultivars and lines of barley from the Far East and Siberia, 15 modern cultivars and breeding lines of barley from Krasnoyarsk and Primorsky Territories, and 11 cultivars approved for cultivation in the northwest of Russia, 14 accessions highly resistant to *Fusarium* blight were identified. Five of them are naked barley forms (k-2946, k-11070, k-11073, k-11076 and k-11082); they yield large grain, but are prone to lodging and susceptible to powdery mildew [112,113]. The genotype of the host plant also has a significant effect on the accumulation of mycotoxins. It was found that there are no cereal crop cultivars immune to *Fusarium* fungi; however, differences in resistance have been observed. Currently, there are several types of resistance to *Fusarium* ear blight (FEB) in cereals: (1) resistance to the penetration of the pathogen; (2) resistance to its dispersal along the ear; (3) seed resistance to infection; (4) tolerance, and (5) the ability to accumulate or degrade toxins [114,115]. The so-called “5th type of resistance” (the ability to accumulate/degrade toxins) affects the final content of toxins and makes it possible to obtain relatively “clean” grain, even under sufficiently high levels of infection. The study of six barley cultivars after artificial inoculation showed that type 4 resistance in barley is associated with type 3 resistance. Resistance to toxin accumulation (called type 5 resistance) is independent of all other types of resistance, while a high β-glucan content in grain was shown to promote type 5 resistance [116].

Thus, the existing global inter- and interspecific, botanical and genetic diversity of oats and barley in the VIR collection is continuously studied for the content of bioactive components in grain, necessary for the development of functional food products.

## 5. Conclusions

Studying plant genetic resources and the processes of biosynthesis of bioactive compounds requires constant improvement of methodological and theoretical approaches. Along with the progress achieved in understanding the role of individual compounds in human life activity, the knowledge about the huge number of compounds serving as sources of functional nutrition required for the normal functioning of the organism is increasing. Therefore, it is necessary to analyze the basic nutritional values of most cereal crops, including oats and barley. By now, the structure and physicochemical properties of cell wall polysaccharides in barley and oat plants have been studied. The main components in the flavonoid formation mechanisms have been discovered, and data on the synthesis of anthocyanins in various tissues of barley have been obtained. However, the enzymes and coding genes responsible for the synthesis of β-glucans and plant polyphenols remain underexplored. Further studies on transcription factors of gene expression and their relationship with other physiological functions of plants will enable breeders to develop new cereals cultivars with desired properties. Such cultivars will be an integral part of the human diet and serve as the basis for food security in every country.

## Figures and Tables

**Figure 1 molecules-26-02260-f001:**
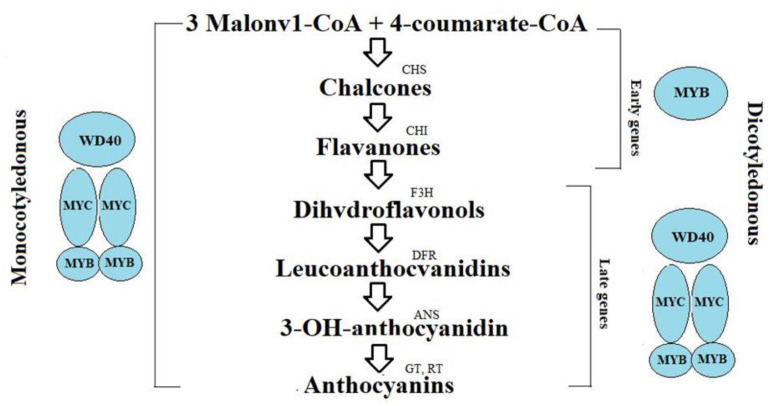
Biosynthesis of plant glycosides and its regulation in monocotyledonous and dicotyledonous plants [54].

**Figure 2 molecules-26-02260-f002:**
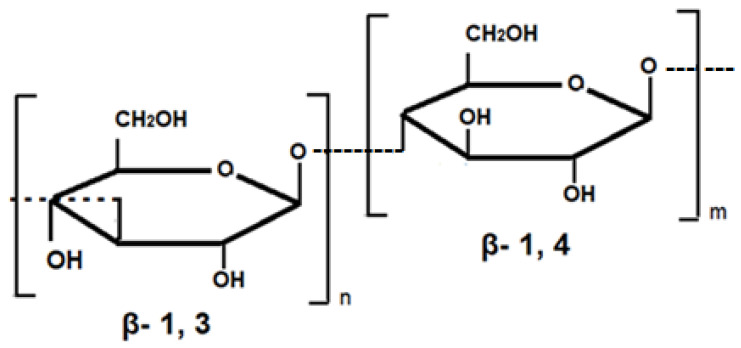
β-Glucans extracted from barley (β-(1,3-1,4)-d-glucan.

**Table 1 molecules-26-02260-t001:** Oil (percent of dry weight seed) and lipid content (percent of total lipid) in wild and cultivated oats samples [34].

Wild/ Cultivated	Oil Content	Pl	1,2-DAG	1,3-DAG	FFA	Unknown Lipids	TAG1	TAG2	TAG
Wild	7.8 ± 0.2	15.4 ± 0.6	1.3 ± 0.1	1.9 ± 0.1	1.5 ± 0.2	1.0 ± 0.1	1.1 ± 0.1	0.9 ± 0.1	77.1 ± 0.9
Cultivated	5.9 ± 0.2	16.0 ± 0.7	1.1 ± 0.1	2.3 ± 0.1	2.5 ± 0.1	1.4 ± 0.1	1.4 ± 0.1	1.1 ± 0.0	74.2 ± 1.0
*F* value	47.5 ***	0.4 NS	1.3 NS	10.6 **	10.2 **	4.2 *	13.2 ***	11.0 **	4.3 *

Abbreviations: PL, polar lipids; 1,2-DAG, 1,2-diacylglycerol; 1,3-DAG, 1,3-diacylglycerol; FFA, free FAs. *F* values are from one-way ANOVA. *** Significant at *p* < 0.001; ** significant at *p* < 0.01; and * significant at *p* < 0.05.

**Table 2 molecules-26-02260-t002:** Content of avenanthramides in oat cultivars [94].

VIR Catalogue No.	Name of Cultivar	Origin	Content of Avenanthramides, mg/kg			
			I	II	III	Average
14787	Privet	RF, Moscow reg.	23.46	36.92	30.39	30.26
15277	Bulanyi	RF, Moscow reg.	8.28	13.06	7.31	9.55
15187	Eklips	RF, Kirov reg.	92.60	121.67	144.32	119.53
14648	Argamak	RF, Kirov reg.	5.38	6.60	6.11	6.03
14857	Krechet	RF, Kirov reg.	20.27	19.91	29.90	23.36
15068	Konkur	RF, Ul’yanovsk reg.	54.32	53.76	50.49	52.86
14960	Vyatskii *	RF, Kirov reg.	214.10	169.50	261.20	214.93
15275	Persheron *	RF, Kirov reg.	62.35	68.82	54.41	61.86
15067	Golets *	RF, Krasnoyask reg.	77.62	82.14	79.24	79.67
15067	Levsha *	RF, Kemerovo reg.	59.81	72.77	67.65	66.74
15115	Aldan *	RF, Kemerovo reg.	60.54	83.74	56.44	66.91
15116	Murom *	RF, Kemerovo reg.	138.98	170.54	200.71	170.08
15117	Pomor *	RF, Kemerovo reg.	43.40	46.82	46.00	45.40
15183	Taidon *	RF, Kemerovo reg.	140.20	165.76	122.96	142.97
14851	Numbat *	Australia	358.87	460.00	403.78	407.55

*—naked cultivars.

**Table 3 molecules-26-02260-t003:** Content of tocopherols and sterols in oat accessions.

VIR Catalogue No.	Name of Cultivar	Origin	Content of Tocopherols, %	Content of Sterols, %
5184	Local	Spain	283	1.18
11840	Borrus	Germany	184	1.00
14648	Argamak	RF, Kirov reg.	189	0.64
13780	Skakun	RF, Moscow reg.	180	0.64
13918	Kirovets	RF, Kirov reg.	227	0.72
13957	Gunter	RF, Kirov reg.	236	0.67
14373	Fakir	RF, Kirov reg.	235	0.81
14781	Faust	RF, Kirov reg.	195	0.77
14857	Krechet	RF, Kirov reg.	149	0.61
15177	Derbi	RF, Ulaynovsk reg.	169	0.62
15180	Piruet	RF, Ulaynovsk reg.	167	0.64
1931	Local *	China	223	0.74
2472	Local *	Mongolia	415	0.97
8317	Local *	China	106	0.85

*—naked cultivars.

**Table 4 molecules-26-02260-t004:** Biochemical characteristics of caryopsis of wild and cultivated oats with different genomes (mg/100 g) [97].

Name/Genomes	Wild Oats						Cultivated Oats
	C	A	AB	AC	CD	ACD	ACD
Amino acids	61.3 ± 0.03	67.5 ± 0.05	45.8 ± 0.02	19.3 ± 0.01	41.1 ± 0.02	30.4 ± 0.02	75.80 ± 0.04
Fatty acids	1058.8 ± 0.11	603.6 ± 0.08	1040.6 ± 0.10	412.0 ± 0.04	656.1 ± 0.07	981.5 ± 0.10	494.00 ± 0.10
Sterols	7.4 ± 0.00	13.6 ± 0.01	25.0 ± 0.01	13.5 ± 0.01	25.7 ± 0.01	26.4 ± 0.02	16.40 ± 0.01
Organic acids	99.6 ± 0.05	126.5 ± 0.08	167.4 ± 0.08	37.1 ± 0.02	115.7 ± 0.06	108.8 ± 0.07	49.90 ± 0.02
Polyhydric alcohols	370.3 ± 0.11	343.2 ± 0.10	342.4 ± 0.10	91.7 ± 0.03	171.3 ± 0.05	312.1 ± 0.09	189.90 ± 0.09
Monosaccharides	1194.6 ± 0.08	1159.1 ± 0.08	1429.5 ± 0.07	358.3 ± 0.02	329.8 ± 0.04	1217.4 ± 0.08	901.50 ± 0.09
Disaccharides	6943.7 ± 0.14	2243.1 ± 0.05	1588.0 ± 0.03	4979.1 ± 0.10	7424.2 ± 0.15	1448.4 ± 0.03	2361.40 ± 0.09
Total sugars	8138.3 ± 0.10	3402.2 ± 0.05	3017.5 ± 0.04	5337.4 ± 0.06	7754.1 ± 0.09	2665.8 ± 0.04	3262.90 ± 0.09

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
