# Peer review of "Bioactive Components in Oat and Barley Grain as a Promising Breeding Trend for Functional Food Production"

_molecules, 2021, doi:10.3390/molecules26082260_

Round 1
Reviewer 1 Report
A well organized review manuscript about the nutritional value of oat and barley grains. Some minor comments that might improve the manuscript: a. in Table 4 it might be more helpful to break down the diploid species to A and C genome species, the tetraploid species to AB and CD genome species and maintain the ACD hexaploids. b. in lines 412-413 you mention about the importance of genotypes and environment in the variability of β-glucans, but most times the interaction of these two is equally or more important, which you don't mention. c. in Table 3 in the two columns showing the content of tocopherols and sterols use mg/kg (you use the symbol % which is not clear to what it refers). and d. when referring to protein content instead of showing %, it would be more informative to show protein yield per hectare, since most times low yields produce higher protein percentages and vice versa.
Author Response
To Reviewer 1
We greatly appreciate the efforts of the reviewer to make our manuscript better. Our response to the reviewer’s comments can be found below and see in text with red.
- in Table 4 it might be more helpful to break down the diploid species to A and C genome species, the tetraploid species to AB and CD genome species and maintain the ACD hexaploids.
А. Table 4 has break down relatively into individual genomes (see text)
- in lines 412-413 you mention about the importance of genotypes and environment in the variability of β-glucans, but most times the interaction of these two is equally or more important, which you don't mention.
А. All biochemical parameters of the caryopsis depend on the growing conditions, but the content of protein, oil and other components, largely depends on the genotype and variety. At the same time, the content of β-glucans is more variable depending on the genotype and variety of the grain, compared to other biochemical parameters. Apparently, the content of β-glucans in naked forms of oats and barley is significantly higher than in hulled varieties. Also, the content of β-glucans in two-row barley varieties is higher than in six-row ones, but the waxy naked grain contains the largest amount of soluble β-glucans, this also depends on the colour of the grain hulls. There is a persistent positive or persistent negative relationship between different indicators of grain and β-glucans, such as oil, yield, nature and weight of 1000 grains (Unpublished data).
- in Table 3 in the two columns showing the content of tocopherols and sterols use mg/kg (you use the symbol % which is not clear to what it refers).
А. It is deleted
- when referring to protein content instead of showing %, it would be more informative to show protein yield per hectare, since most times low yields produce higher protein percentages and vice versa.
А. You are absolutely right. It is added
High productive with high-protein content forms were identified among both naked and hulled accessions of barley [110].
Reviewer 2 Report
Review of Bioactive Components in Oat and Barley Grain
Academician Dr. Aleksei Konarev is to be congratulated for assembling such a writers’ consortium capable of producing such a masterful assessment of cereal functionality in our diet.
The review of the field is strong, comprehensive, and engaging for it portrays the primal position in human civilization that is the Gramineae. It is not so much that humans conscientiously chose to husband the grasses and small grains but it turns out that they themselves with their active ingredients culture and nurture our gut microbiota and supply us with conduits for the gut-brain axis. Dr. Konarev cites 8 or 9 of his own papers out of the 117 or so citations in the bibliography, so this is justified given his life long devotion to cereal science, and not only his but that of his father also. A legacy almost as strong as Vavilov himself. It is also good to see inclusion of the pioneering work reported by Peter Wood and Dave Paton of Agriculture Canada. Speaking of which, AAFC, the authors could consider adding the classical case of the pain killer Tylenol being found naturally in oat, as a startling example of a real active ingredient in oat: High Levels of Avenanthramides in Oat-Based Diet Further Suppress High Fat Diet-Induced Atherosclerosis in LdIr(-/-) Mice, Thomas, Michael; Kim, Sharon; Guo, Weimin; et al. JOURNAL OF AGRICULTURAL AND FOOD CHEMISTRY 66(2): 498-504 2018 DOI: 10.1021/acs.jafc.7b04860
Overall the manuscript is well structured and well presented. However, the authors do need to make more effort to correct the typography especially when citing other authors like Peter Shewry. Ref # 25 has Shewrt. It is not the job of the editor or reviewer to ensure that accuracy and precision represented by this manuscript are at the highest level possible. What else might keep the International Space Station still in orbit over our heads if it is not accuracy and precision? Some other specific comments follow below.
Abstract
Line 18: VIR – acronym not previously spelled out. In the text it should first be presented as full prose, “NI Vavilov Institute of Plant Industry (VIR)”. The world knows full well who was our hero Dr. Vavilov, but the research institute’s name is not as familiar to global science. Cereal nutrition science is an international venture and enterprise.
Introduction
Line 29: whose – incorrect usage.
Lines 27-30: no reference for the statement of reduced risk of metabolic diseases. Please include.
Main text
Table 1: Did authors do their own analysis of data or acquired the data and table from Leonova et al. 2008?
Figure 1: Would it be possible to increase resolution of Figure 1?
Figure 2: Not referenced in text. Are the Figure components created by authors? Do the images have a creative commons license, if needed?
Overall
Minor sentence grammar errors. A professional English writer should improve the flow and feel of the text once its scientific accuracy is assessed to level of triple redundancy. E.g. not all years are bolded, e.g. #26 has 2011.
An informative summary of bioactive components in barley and oat and their relevance in food production. With these minor revisions this article will be a useful and guiding beacon for others.
Author Response
To Reviewer 2
We greatly appreciate the efforts of the reviewer to make our manuscript better. Our response to the reviewer’s comments can be found below and see in text with red.
Q. Speaking of which, AAFC, the authors could consider adding the classical case of the pain killer Tylenol being found naturally in oat, as a startling example of a real active ingredient in oat: High Levels of Avenanthramides in Oat-Based Diet Further Suppress High Fat Diet-Induced Atherosclerosis in LdIr(-/-) Mice, Thomas, Michael; Kim, Sharon; Guo, Weimin; et al. JOURNAL OF AGRICULTURAL AND FOOD CHEMISTRY 66(2): 498-504 2018 DOI: 10.1021/acs.jafc.7b04860
A. It is added to the text and Reference.
Oat grains contain unique antioxidants - avenanthramides, which inhibit inflammatory processes in endothelial cells. It had shown that oat-based diets in mice reduce the risk of atherosclerosis. The processes of lipoes distribution and the becoming of atherosclerosis in mice and humans are liking, therefore oat avenanthramides are very important for the prevention of human cardiovascular diseases (Thomas, et al 2018)
Thomas M.; Kim S.; Guo W.; Collins W.F.; Wise M.L.; Meydani M. High levels of avenanthramides in oat-based diet further suppress high fat diet-induced atherosclerosis in Ldlr−/− Mice. J. Agricultural and food chemistry 2018, 66(2), 498-504, doi:10.1021/acs.jafc.7b04860
Q. Overall the manuscript is well structured and well presented. However, the authors do need to make more effort to correct the typography especially when citing other authors like Peter Shewry. Ref # 25 has Shewrt. It is not the job of the editor or reviewer to ensure that accuracy and precision represented by this manuscript are at the highest level possible.
A. It is changed, see Reference.
Line 18: VIR – acronym not previously spelled out. In the text it should first be presented as full prose, “NI Vavilov Institute of Plant Industry (VIR)”. The world knows full well who was our hero Dr. Vavilov, but the research institute’s name is not as familiar to global science. Cereal nutrition science is an international venture and enterprise.
А. It is added to Abstract and text.
Line 29: whose – incorrect usage.
A. It is changed to which
Lines 27-30: no reference for the statement of reduced risk of metabolic diseases. Please include.
A.It is added
Nikberg I.I. Functional products in the structure of modern nutrition. Int. J. Endocrinology 2011, 6, 64-69. (in Russian)
Table 1: Did authors do their own analysis of data or acquired the data and table from Leonova et al. 2008?
А. It is added
Table 1. Oil (percent of dry weight seed) and lipid content (percent of total lipid) in wild and cultivated oats samples [34]
Figure 1: Would it be possible to increase resolution of Figure 1?
А. It is changed (see text)
Figure 2: Not referenced in text. Are the Figure components created by authors? Do the images have a creative commons license, if needed?
А. It is changed (see text)
Overall
Minor sentence grammar errors. A professional English writer should improve the flow and feel of the text once its scientific accuracy is assessed to level of triple redundancy. E.g. not all years are bolded, e.g. #26 has 2011.
А. It is changed (see text and Reference)